# Developing Automated Computer Algorithms to Track Periodontal Disease Change from Longitudinal Electronic Dental Records

**DOI:** 10.3390/diagnostics13061028

**Published:** 2023-03-08

**Authors:** Jay S. Patel, Krishna Kumar, Ahad Zai, Daniel Shin, Lisa Willis, Thankam P. Thyvalikakath

**Affiliations:** 1Dental Informatics, Department of Cariology Operative Dentistry and Dental Public Health, Indiana Univesity School of Dentistry, Indianapolis, IN 46202, USA; 2Health Informatics, Department of Health Services Administrations and Policy, Temple University College of Public Health, Philadelphia, PA 19122, USA; 3Department of Oral Health Sciences, Temple University Kornberg School of Dentistry, Philadelphia, PA 19140, USA; 4Dental Informatics Program, Center for Biomedical Informatics, Regenstrief Institute, Indianapolis, IN 46202, USA

**Keywords:** periodontal disease, electronic dental record, longitudinal data, data quality, dental informatics, clinical course of periodontal disease, periodontal cohort generation

## Abstract

Objective: To develop two automated computer algorithms to extract information from clinical notes, and to generate three cohorts of patients (disease improvement, disease progression, and no disease change) to track periodontal disease (PD) change over time using longitudinal electronic dental records (EDR). Methods: We conducted a retrospective study of 28,908 patients who received a comprehensive oral evaluation between 1 January 2009, and 31 December 2014, at Indiana University School of Dentistry (IUSD) clinics. We utilized various Python libraries, such as Pandas, TensorFlow, and PyTorch, and a natural language tool kit to develop and test computer algorithms. We tested the performance through a manual review process by generating a confusion matrix. We calculated precision, recall, sensitivity, specificity, and accuracy to evaluate the performances of the algorithms. Finally, we evaluated the density of longitudinal EDR data for the following follow-up times: (1) None; (2) Up to 5 years; (3) > 5 and ≤ 10 years; and (4) >10 and ≤ 15 years. Results: Thirty-four percent (*n* = 9954) of the study cohort had up to five years of follow-up visits, with an average of 2.78 visits with periodontal charting information. For clinician-documented diagnoses from clinical notes, 42% of patients (*n* = 5562) had at least two PD diagnoses to determine their disease change. In this cohort, with clinician-documented diagnoses, 72% percent of patients (*n* = 3919) did not have a disease status change between their first and last visits, 669 (13%) patients’ disease status progressed, and 589 (11%) patients’ disease improved. Conclusions: This study demonstrated the feasibility of utilizing longitudinal EDR data to track disease changes over 15 years during the observation study period. We provided detailed steps and computer algorithms to clean and preprocess the EDR data and generated three cohorts of patients. This information can now be utilized for studying clinical courses using artificial intelligence and machine learning methods.

## 1. Introduction

Despite advances in periodontal disease (PD) research and treatments, nearly 42% of adults in the United States (US) suffer from PD [1]. If PD is left untreated, it can lead to tooth loss and poor quality of life [2]. Research has demonstrated that PD can be prevented if the associated risk factors are controlled [3,4,5]. For example, some studies have followed patients over time to evaluate the long-term effect of smoking, diabetes, age, and calculus on PD initiation and progression [6,7,8,9,10]. While these studies provided meaningful insights, most study cohorts were from 1969 to 1988, which may not represent the current patient population [6,11,12]. In addition, these studies were conducted on the non-US patient population and did not have longer follow-up visit information [6]. Observing disease change over time is necessary, especially for PD, which is a slow-progressing disease [13]. However, it is difficult to conduct such studies because it is expensive, laborious, time-consuming, and difficult to retain patients for a long time [14].

The high usage of electronic dental record (EDR) systems to document patient care information provides a huge opportunity to study the clinical course of PD, as well as the influence of risk factors [15,16,17]. The EDR has many advantages for conducting longitudinal studies. For example, the EDR can provide a longer follow-up study period, provide patients’ up-to-date clinical information, and provide real-world evidence [18,19,20]. Despite this promising potential, EDR data have challenges, such as questionable quality, fragmented information documented in different sections of the EDR, and missing information. For example, to study the clinical course of disease using EDR data, the *t*0 (no disease state) progressing to *t*1…*tn* (disease stage) is critical. However, many patients visit dentists when the disease is already in the active stage. As a result, it is unknown whether EDR data has the potential to provide patients with important *t*0 information [21]. Moreover, methods such as machine learning (ML) and artificial intelligence (AI) can be applied to these datasets to predict the risk of disease initiation and progression. However, to apply these methods, evaluating the quality of the data is critical to avoid flawed outcomes.

Several studies have utilized longitudinal electronic health record (EHR) data to predict hospitalization rates and risk of cardiovascular disease in medicine [22,23]. Similarly, in dentistry, researchers have utilized automated approaches to compare the completeness of periodontal charting information in four large US dental academic institutes’ EDR data and automated diagnosis (using SQL) [24]. The authors also determined new periodontitis cases and tooth loss by leveraging EDR data from three of these institutions [25]. Another study reported a deep-learning model that automated the staging and grading of periodontitis [26]. Finally, a few studies have reported methods to extract PD risk factors information, such as smoking, diabetes, and cardiovascular diseases from the EDR [27,28,29,30]. Yet, to the best of our knowledge, no study has utilized longitudinal EDR data to study PD change and its clinical course over time.

Therefore, our *long-term goal* is to utilize longitudinal EDR data to examine the clinical course of PD using AI and ML methods and assess long-term treatment outcomes of surgical and non-surgical periodontal treatments. The objective of this study was to develop two automated algorithms to track patients’ PD progression over time and to determine the quality of the longitudinal EDR data. We generated three patient cohorts: (1) Disease progression; (2) Disease improvement; and (3) No disease change. We provided detailed steps and computer algorithms to clean and pre-process the messy EDR data and generated three cohorts of patients. These open-source resources can be utilized by other researchers for their PD-related research. Finally, the automated algorithms developed in this study are made publicly available for use by other researchers.

## 2. Materials and Methods

Figure 1 demonstrates the overall workflow of this study. We first extract patients’ PD-related information such as diagnosis, charting, and date of diagnosis from the EDR. Patients’ PD diagnosis information is typically documented in the free-text format. Therefore, we first created and tested NLP algorithm to extract this information in a structured format. Next, we created a second automated algorithm to classify patients into three groups (disease progression, disease improvement, no change in the disease) using this PD diagnosis information in a structured format. We evaluated the performance of the algorithms using evaluations measures such as precision, recall, f-1 score, and accuracy. Finally, we applied these algorithms on the 15-year longitudinal dataset to generate final patient cohorts.

### 2.1. Data Source

We utilized EDR (axiUm^®^, Exan software, Las Vegas, NV, USA) data from Indiana University School of Dentistry (IUSD) predoctoral clinics to conduct this study. The data included periodontal examination findings (charting) through clinical periodontal notes of patients who underwent comprehensive oral examination (COE) between 1 January 2009, and 31 December 2014, and who were 18 years or older during their first completed COE during this period. The patients’ visit information that may fall outside this period was also included in this study. For example, if the patient received COE in 2010 and received treatments in 2007 and 2015, information from 2007 and 2015 would also be included. We excluded edentulous patients and patients who only had paper records.

### 2.2. Natural Language Processing (NLP) Algorithm (PD Extractor.py) to Extract PD Diagnoses from Periodontal Evaluation Forms

We developed an NLP algorithm *PD Extractor.py* to retrieve PD diagnosis written as free text in the periodontal evaluation form. Clinicians typically write PD type (gingivitis or periodontitis), severity (mild, mild to moderate, moderate, moderate to severe, and severe), location (maxilla, mandible, tooth number), onset (acute or chronic), and extent (localized or generalized). The *PD Extractor.py* was developed using a bottom-up approach. Two expert clinicians first developed annotation guidelines by manually reviewing 50 patient clinical notes to understand the writing pattern of PD diagnosis in the EDR clinical notes. We then reviewed a total of 200 additional notes and manually annotated various writing PD diagnoses. The disagreements between the reviewers were resolved through consensuses, and a gold-standard dataset was created. This gold-standard dataset was then split into training and testing sets to train and test a named entity recognition (NER) NLP algorithm. We utilized the approximate string-matching function (ASM) and Levestein Distant Function to develop the NLP algorithm. Details on each Python function utilized, and the full source code for the NLP algorithm, are described elsewhere [31]. During the processing of this step, we encountered one limitation.

Not all records contained patients’ detailed PD information, such as PD type, severity, location, onset, and extent. Therefore, we used the stepwise bottom-up processing approach, as demonstrated in Figure 2, in which the application would be considered as maximum information detail. However, if any detailed structured categories are missing, the *PD Extractor.py* will not throw an error and extract the limited information present in the clinical note. For example, suppose the clinical note contains “mild periodontitis” and does not contain information on the location or extension. In that case, the program will still extract this information in a structured format and leave empty categories, such as location, onset, and extent.

### 2.3. A computer Algorithm (PD Change Classifier.py) That Automatically Determines PD Change Overtime

Lastly, we developed a computer algorithm PD Change Classifier.py that examines patients’ PD diagnoses information in each consecutive visit and classifies it into one of the following categories:PD progression: e.g., from mild gingivitis to mild periodontitis, from mild periodontitis to moderate periodontitis, etc.No change in disease status: e.g., from mild gingivitis to mild gingivitis, from mild periodontitis to mild periodontitis, etc.Disease improvement: e.g., from moderate periodontitis to mild periodontitis, from severe periodontitis to mild periodontitis, etc.

*PD Change Classifier.py* application consisted of several Python libraries that include Natural Library Toolkit (NLTK), string, regular expression, and Pandas. By using these libraries, first, the *PD Change Classifier.py* read the text file and saved disease type, severity, and disease extent in temporary variables. Next, the classifier created two temporary variables, “From” and “To,” and determined the date difference between the two visit dates. If these two dates were different (differences have to be 90 days apart), then the diagnosis from the first date was placed in the “From” temporary variable. Similarly, the diagnosis recorded at the latest date was placed in the “To” temporary variable. Next, it determined if these two dates recorded in the “From” and “To” variables were similar or not. The application skipped these records and went to the next available date if they were identical. If there was no other diagnosis present, it went to the next row (patient ID). Figure 3 demonstrates an example of the output of the *PD Change Classifier.py.*

### 2.4. Evaluate the Performance of Automated Computer Applications

Two clinical faculty manually reviewed 250 PD-diagnosed patients using the PD diagnosis categories reported in a previous US population PD prevalence study [1]. Similarly, they also reviewed 250 clinical notes containing patients’ PD diagnoses. The final inter-rater agreement was one that demonstrated excellent agreement (Cohen’s Kappa = 1). These manually reviewed datasets were compared against the computer-generated outputs. Next, a confusion matrix containing true positive (TP), false positive (FP), true negative (TN), and false negative (FN) were created for both algorithms [32]. Using this confusion matrix, we calculated precision (correctly predicted positive observations to the total predicted positive observations), recall (correctly predicted positive observations to all observations in actual class), and F-1 measure (weighted average of Precision and Recall) to assess performances [33].

### 2.5. Observation Time and Dentisty of the Longituidional EDR Data

Descriptive statistics with 95% confidential intervals were performed on the number of periodontal charting and clinician-recorded diagnoses documented between 1 June 2005, and 1 August 2019 for the patients who received at least one COE between 1 January 2009, and 31 December 2014. The rationale for including periodontal charting information along with clinician-documented diagnosis is for comparison purposes. We wanted to compare the periodontal charting documentation versus clinician-documented diagnoses. Moreover, in our study (Patel et al. 2023), we have developed automated approaches to diagnose patients’ periodontitis status into a PD classification recommended by the American Academy of Periodontology (AAP) [1]. The average days, months, and years between patients’ first, and second; first and third; and first and fourth visits were calculated. This test helped us identify how frequently patients’ clinician-recorded diagnoses were available to determine their disease change over time. The frequency count and the number of patients by the observation time between their first and last visits were generated. The frequency counts were generated in the following six categories: (1) No-follow-ups; (2) Up to 5 years; (3) >5 and ≤ 10 years; (4) >10 and ≤ 15 years; (5) > 15 and ≤ 20 years; and (6) more than 20 years.

### 2.6. Data Analysis

Descriptive statistics with 95% confidential intervals were performed on the clinician-documented diagnoses between 1 June 2005, and 1 August 2019. The average number of days, months, and years between patients’ first and second; first and third; and first and fourth visits was calculated. This test helped us identify how frequently patients’ clinician-documented diagnoses were available to determine their disease change over time. The frequency count and the number of patients by the observation time between their first and last visits were generated. The frequency counts were generated in the following four follow-up categories: (1) None; (2) Up to 5 years; (3) > 5 and ≤ 10 years; and (4) > 10 and ≤ 15 years. Last, the frequency count of the number of patients whose disease status did not change, disease status progressed, and disease status improved from their first to the last visit using patients’ clinician-recorded diagnoses was also generated.

## 3. Results

### 3.1. Patient Demographics

The EDR data included 28,908 distinct patients who received at least one COE between 1 January 2009, and 31 December 2014. Fifty-four percent of patients were females with a mean age of 46 years (standard deviation = 16.74). Seventy-nine percent (N = 22,880) of patients had at least one full-mouth periodontal finding, and 13,219 patients had both clinician-documented diagnoses in the EDR.

### 3.2. Performances of the Two Automated Algorithms

PD Extractor.py achieved 98% accuracy, as demonstrated in Table 1. We have provided detailed descriptions of the reasons for an excellent performance, as well as an error analysis of the manual review in a previous publication (Patel et al. 2020). This publication also provides examples of the free-text information document in our EDR. Similarly, the PD Change Classifier.py provided excellent results (accuracy of 98%) as demonstrated in Table 2 in assigning patients into one of the three cohorts (disease improvement, disease progression, and no disease change). This is because structured information extracted from PD Exctractor.py was used to determine the disease change. Hence, this algorithm mainly included rule-based logics, which were studied to provide excellent results. Using this algorithm, we were able to automatically classify patients in one of the three groups by evaluating their 15 years of longitudinal EDR data.

### 3.3. Periodontitis Cases Automatically Classified by Periodontitis_Diagnoser.py and PD Extractor.py

Table 3 demonstrates the automated generated diagnosis using the NLP program from periodontal clinical notes. The clinician-documented diagnoses were available for 13,219 patients (46%). Among these patients, 3193 patients (24%) were diagnosed with mild gingivitis, 1607 (12%) with moderate gingivitis, and 143 (1%) with severe gingivitis out of 13,219 available periodontal evaluation forms. Eighteen percent of patients (2430) were diagnosed with mild periodontitis, 1899 (14%) with moderate periodontitis, and 554 (4%) with severe periodontitis cases.

### 3.4. Observation Time of Longitudinal EDR Data

There were 63,552 periodontal charts documented for 22,880 unique patients. The observation time of patients who had at least one periodontal charting dataset ranged from 0 to 15 years. A total of 15,217 (53%) patients out of 28,908 (100%) had no follow-up visits, 9954 (34%) patients had up to 5 years of observation time, 3203 (11%) had 5 years to 10 years of observation time, and 534 (2%) patients had 10 years to 15 years of observation time. For the periodontal charting findings, the average visit was 2.78 (median = 2, standard deviation = 2.9) (see Table 4).

There were 20,152 clinician-documented diagnoses for 13,219 unique patients. The average documented PD diagnosis was 1.52 (median = 1, standard deviation = 1) for 13,114 unique patients. We found that 7657 (58%) patients had exclusively one clinician-documented PD diagnosis, 3197 (24%) had exclusively two diagnoses, 1052 (8%) had three diagnoses, and 1313 (10%) patients had 4 to 28 PD diagnoses. There were 5562 patients who had more than one clinician-documented diagnosis available to determine their disease change between their first and last visits (see Table 5).

Among the 5562 patients who had more than two diagnoses available, the average time period between their first and second visit was 0.9 years (approximately 11 months [346 days]) (standard deviation of 584 days); the first and third visit was 1.6 years (approximately 19 months [588 days]) (standard deviation of 709 days); and first and fourth visit was 3 years (approximately 35 months [1072 days]) (standard deviation of 855 days).

### 3.5. Number of Patients Whose Periodontal Diagnosis Changed over Time

When considering clinician-documented diagnoses, 72% percent of patients (n = 3919) out of 5562 (100%) did not have a disease status change between their first and last visits. See Appendix A for detailed categories.

We found 669 (13%) patients’ disease status progressed between their first and last visit. The top three categories in disease progression include the following:Seventy-seven (12%) out of 669 (100%) patients progressed from generalized mild periodontitis to localized moderate periodontitis.Sixty-six (10%) progressed from generalized moderate periodontitis to localized severe periodontitis.Fifty-six (9%) progressed from generalized mild periodontitis to generalized moderate periodontitis. See Appendix A for detailed categories.

There were 589 (11%) patients out of 5562 (100%) patients whose disease improved between their first and last visits. The top three categories in disease improvement included:Seventy-six (13%) out of 537 (100%) patients progressed from generalized moderate periodontitis to generalized mild periodontitis.Thirty-two (5%) progressed from generalized mild periodontitis to generalized mild gingivitis.Thirty (5%) progressed from generalized mild periodontitis to localized mild periodontitis. See Appendix A for detailed categories.

There were 437 (7%) patients out of 5486 (100%) patients in the unknown category. See Appendix A for detailed categories.

### 3.6. Performance of the Automated Applications

As demonstrated in the article [31], we achieved excellent results with 99% precision, 100% recall, and 99.5% F-measure for the *Periodontitis_Diagnoser.py,* and an average of 98% precision, recall, and F-measure of the *PD Extractor.py*. For the PD Change *Classifier.py* application, we achieved excellent results with 97% precision, 99% recall, and 98% F-measure.

## 4. Discussion

This retrospective study demonstrated the feasibility of using longitudinal EDR data to track changes in PD diagnosis and determined the quality of the longitudinal EDR data for clinical research. We found 34% of our patients (*n* = 9954) had up to five years of follow-up visits with an average visit of 2.78 when their periodontal charting information was utilized. We found an average of three patient visits per year when periodontal charts (63,552) were utilized to obtain their periodontal diagnosis. Similarly, when clinician-documented diagnoses were considered, we found 42% of patients (*n* = 5562) who had at least two PD diagnoses to determine the disease change. Moreover, we were able to successfully classify the cohort of patients whose disease statuses changed over time. This diagnosis information with patients’ other medical history, dental history, and social history would allow us to study the clinical course of PD.

### 4.1. No Disease Change Group

Our automated application *PD Change Classifier.py* determined 72% of patients (*n* = 3919) fell into the “no disease change” category between their first and last dental appointments. We believe that the patients falling in this category may have received periodontal treatment, oral prophylaxis, and preventive treatments regularly. As a result, the treatments provided at the right interval would have prevented the disease progression in these patients. Further studies determining treatment outcomes among these patients would allow us to determine the effectiveness of periodontal treatments. It is also interesting to note that most of these patients’ disease stages were still mild–moderate periodontitis cases. This provides us some insight: If PD can be diagnosed early, we can retain patients for the long term in milder PD stages and prevent tooth loss. More studies are essential to examine the effectiveness of periodontal treatments in mild–moderate periodontitis cases. One other reason for “no disease change” could be because patient visits were clustered close to their initial COE date. For example, if the patient had 10 years of follow-up, most visits were clustered either in the beginning or later period. This may not provide the complete picture of disease progress over the 10 longitudinal years.

### 4.2. Disease Progression Group

The disease progression group included 669 (13%) patients whose disease status progressed between their first and last visit. The progression could be due to various reasons that require further investigation. Risk factors such as smoking, diabetes, and other common inflammatory systemic diseases may contribute to the patients’ disease progression. Next, home-based oral healthcare and compliance are major contributors to the success of periodontal treatments. These patients may or may not be compliant with the home hygiene instruction, and their disease could have progressed. Therefore, future studies should also evaluate the influence patient compliance on their oral health.

### 4.3. Disease Improvement Group

In the disease improvement group, 589 (11%) patients’ disease status improved between their first and last visits. Many reasons may have improved these patients’ disease status. First, most of the improved disease categories were mild-to-moderate periodontitis. Typically, when patients are on long-term periodontal maintenance therapy, their periodontal pocket depth and clinical attachment are expected to improve over time. Next, research studies have demonstrated that if periodontitis is diagnosed and treated in early stages (mild to moderate), then a good prognosis can be achieved, and the patient can remain progression-free [34]. Last, the maximum improvement was observed in the extent of periodontitis. For example, many patients who had generalized periodontitis improved to localized periodontitis after receiving periodontal treatments. Limitations & Future work: Like any study, we encountered some limitations. First, these study results may not be generalizable because they included EDR data only from one institution. Nevertheless, this study demonstrated a step-by-step approach to evaluate the quality of longitudinal EDR data that future studies can adopt and expand further. To facilitate further studies, the computational programs are shared through this paper. Next, the NLP algorithm used in this study may or may not work optimally on other datasets due to variations in documentation across different institutions. Nonetheless, researchers could use the basic NLP framework as demonstrated in this paper to retrieve their clinician-documented diagnoses. In the future, we will utilize these cohort of patients to develop prediction models using AI and ML to determine the clinical course of PD and examine long-term PD treatment outcomes.

## 5. Conclusions

This study developed two automated computer algorithms to classify patients into three groups (disease progression, disease improvement, no change in disease) with 99% accuracy. We also demonstrated a step-by-step process to clean and process messy EDR data to extract information from free text, which is essential before utilizing AI and ML methods. We also demonstrated the feasibility of utilizing longitudinal EDR data to track the disease change over 15 years. We successfully generated three different cohorts of PD patients (no disease change, disease progression, disease improvement) to study the clinical course of PD. This approach can be used to investigate longitudinal EDR data for PD response to different treatments.

## Figures and Tables

**Figure 1 diagnostics-13-01028-f001:**
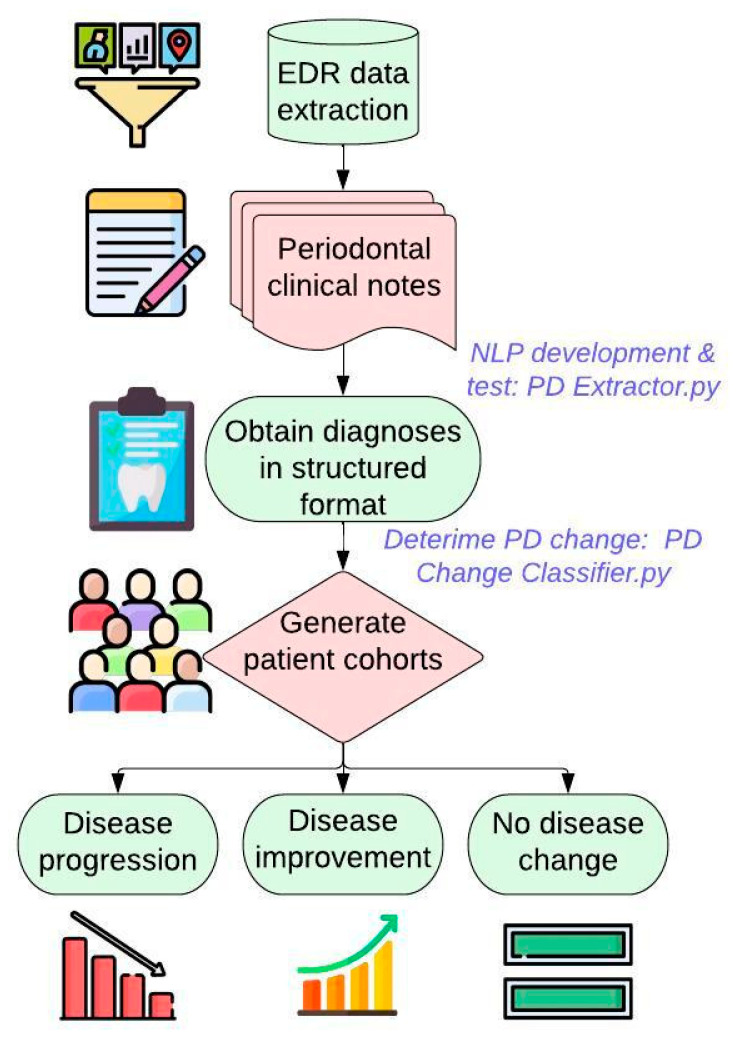
Developing automated computer algorithms to generate three patient cohorts.

**Figure 2 diagnostics-13-01028-f002:**
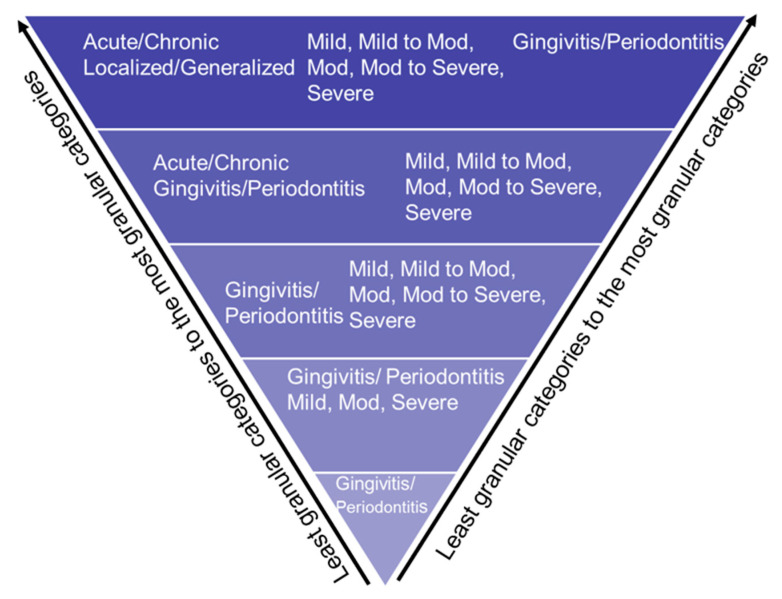
Bottom-up approach to extract PD based on disease type, disease severity, disease location, and disease extension.

**Figure 3 diagnostics-13-01028-f003:**
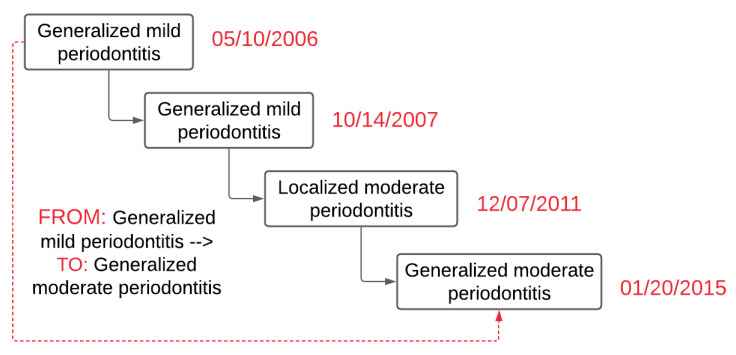
Example output of diagnosis change overtime classifier.py to determine PD change over time (a hypothetical case).

**Table 1 diagnostics-13-01028-t001:** Performance of the PD Extractor.py.

Performance Measures	Value (%)
Sensitivity	98
Specificity	97
Precision	98
Positive predictive value	98
Negative predictive value	97
Accuracy	98
Matthews Correlation Coefficient	96

**Table 2 diagnostics-13-01028-t002:** Performance of the PD Change Classifier.py.

Performance Measures	Value (%)
Sensitivity	99
Specificity	98
Precision	98
Positive predictive value	98
Negative predictive value	99
Accuracy	98
Matthews Correlation Coefficient	97

**Table 3 diagnostics-13-01028-t003:** Periodontal diagnoses generated from clinical notes.

Diagnoses Generated from Clinical Notes		
Mild gingivitis	3193	(24)
Mild-to-moderate gingivitis	247	(2)
Moderate gingivitis	1607	(12)
Moderate-to-severe gingivitis	62	(0.5)
Gingivitis	1613	(12)
Severe gingivitis	143	(1)
Mild periodontitis	2430	(18)
Mild-to-moderate periodontitis	569	(4)
Moderate periodontitis	1899	(14)
Moderate-to-severe periodontitis	350	(3)
Periodontitis	258	(2)
Severe periodontitis	554	(4)
Missing/no disease mentioned/algorithm error	294	(2)
Total (available data)	13,219	(100)
Missing data	15,689	(54)
**Total**	**28,908**	**(100)**

**Table 4 diagnostics-13-01028-t004:** Number of patients by the observation time between the first and last visits, from 1 June 2005, to 1 August 2019, while using periodontal charts.

Time in Years (Observation Time)	N	(%)
No follow-up	15,217	(53)
Up to 5 years	9954	(34)
>5 and ≤ 10 years	3203	(11)
>10 and ≤ 15 years	534	(2)
Total	28,908	(100)

**Table 5 diagnostics-13-01028-t005:** Number of patients by the observation time between the first and last visits, from 1 June 2005, to 1 August 2019, while using periodontal clinical notes (clinician-documented diagnoses).

Time in Years (Observation Time)	Frequency	(%)
No follow-up	10,521	(37)
Up to 5 years	9651	(33)
>5 and ≤ 10 years	2322	(8)
>10 and ≤ 15 years	386	(1)
>15 and ≤ 20 years	0	(0)
Missing data	6028	(21)
Total	28,908	(100)

## Data Availability

The dataset for this study contains identifiable information and therefore, regulatory policies do not grant permission to share publicly. Data can be shared upon request after appropriate institutional review and approval and agreements for research and data sharing are executed.

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
