# Peer review of "Developing Automated Computer Algorithms to Track Periodontal Disease Change from Longitudinal Electronic Dental Records"

_diagnostics, 2023, doi:10.3390/diagnostics13061028_

Round 1

Reviewer 1 Report

The authors have done a wonderful job in conducting a feasibility study to track changes in periodontal disease status from electronic dental records generated at the Indian University School of Dentistry. The manuscript is nicely written, the concepts well-described, and the data/statistical analysis conducted satisfactorily. I have not much to comment, other than considering the ones  below: 

(a) I see several inconsistency in the writing, in particular, the first paragraph of the Introduction. Care must be taken to fix those, and also throughout the manuscript. 

(b) The authors state that the NLP programs used are shared (Discussion section, last para); how are they shared? 

Author Response

Note: Dear reviewer, I request you to see the attached word document for better visibility.

Response to the Reviewers' Feedback

We want to thank all reviewers and editors for providing their critical feedback and suggestions, which have helped us improve our manuscript tremendously. We want to note that we spent a significant amount of time addressing reviewers' comments especially making the study focus technical to fit the publication in the special issue. We have strengthened the rationale of evaluating the quality of longitudinal data and generating a cohort of patients automatically in the updated manuscript. We have provided detailed descriptions of the two automated algorithms developed in this study to clean and preprocess electronic dental record (EDR) data for research. Finally, we addressed all 24 unique comments (100%) provided by the reviewers (n=4). We have made all changes in track change; therefore, please ensure to have the track changes correspond to the accurate line and page numbers. We hope you find these edits helpful and meet your expectations.

Reviewer 1

1. The authors have done a wonderful job in conducting a feasibility study to track changes in periodontal disease status from electronic dental records generated at the Indian University School of Dentistry. The manuscript is nicely written, the concepts well-described, and the data/statistical analysis conducted satisfactorily. I have not much to comment, other than considering the ones below:

Response: Thank you, we appreciate your positive feedback.

2. I see several inconsistency in the writing, in particular, the first paragraph of the Introduction. Care must be taken to fix those, and also throughout the manuscript.

Response: Thank you for catching this error. We have now made the language consistent across the manuscript. The updated manuscript is now significantly improved.

3. The authors state that the NLP programs used are shared (Discussion section, last para); how are they shared?

Response: Thank you for this comment. We have shared our computer algorithms that automatically diagnose periodontal disease and NLP applications that extract information from free-text clinical notes in the supplemental material. Please see the supplemental materials. We have also provided access to GitHub and source codes for public use. I hope this increases the research reproducibility.

Reviewer 2 Report

I have carefully read the article proposed by the authors. I totally agree on the fact of being interested in periodontitis, a public health problem. However, I have great difficulty in seeing the interest of the proposed solution.

1 - since periodontitis is a multifactorial pathology with modifiable and non-modifiable risk factors. Several risk factors should be mentioned in the introduction, whether they are local, systemic, linked to the individual, to society, biological, socio-demographic... On the other hand, the system proposed by the authors could have been all the more useful as it would have extracted this information from the EDR.

2 - I don't understand at all why NLP is necessary since the authors have periodontal charting. Please justifiy and detail.

3 - Are the authors using the last periodontal classification ?

4 - What does mean NLP ? If this article presents a data processing flow, this step must be detailed. Is the NLP based on deep learning ? (in such detail the network structure, dataset training, validation and test) or regular expression (detail the structure of the regexp and its accuracy).

5 - Where is the python code ?

6 - I would also have expected the article to focus on the prediction of the evolution of periodontitis from the initial diagnosis and charting.

In the end this article contains very interesting ideas, but from a computer science point of view it is not sufficient (lack of details, lack of implementation) and from a medical point of view it does not bring any tangible element on periodontitis.

Author Response

Note: Dear reviewer, I request you to see the attached word document for better visibility.

Response to the Reviewers' Feedback

We want to thank all reviewers and editors for providing their critical feedback and suggestions, which have helped us improve our manuscript tremendously. We want to note that we spent a significant amount of time addressing reviewers' comments especially making the study focus technical to fit the publication in the special issue. We have strengthened the rationale of evaluating the quality of longitudinal data and generating a cohort of patients automatically in the updated manuscript. We have provided detailed descriptions of the two automated algorithms developed in this study to clean and preprocess electronic dental record (EDR) data for research. Finally, we addressed all 24 unique comments (100%) provided by the reviewers (n=4). We have also shared our computer algorithms that automatically diagnose periodontal disease and NLP applications that extract information from free-text clinical notes in the supplemental material. We have also provided access to GitHub and source codes for public use. I hope this increases the research reproducibility. We have made all changes in track change; therefore, please ensure to have the track changes correspond to the accurate line and page numbers. We hope you find these edits helpful and meet your expectations.

Reviewer 2

1. I have carefully read the article proposed by the authors. I totally agree on the fact of being interested in periodontitis, a public health problem. However, I have great difficulty in seeing the interest of the proposed solution.

Response: We appreciate you see an interest in studying periodontitis. Below we are attempting to provide interest in the proposed solution. We have also changed this manuscript significantly especially adding technical components. The title, abstract, and main text of the manuscript has been changed as per your suggestion. As an expert, you must be aware that most of the health information technologies, especially prediction models developed using poor-quality data, may not work in a real-world setting. Therefore, ample literature has recently demonstrated the importance of evaluating the quality of the electronic health record (EHR) data before its intended use for research to avoid the "garbage in, garbage out" situation (cited in the main manuscript). In this study, we attempted to first determine the quality of the longitudinal electronic dental record (EDR) data to examine if this data can be utilized to study periodontal disease (PD). Moreover, the EDR data is messy and automated approaches are needed to utilize the data for research purposes. You may have found our research question simple; however, we developed two complex computational approaches to automate diagnosis. Any researchers worldwide would be able to utilize these open-source computer algorithms to improve the completeness of PD diagnosis documentation. This information is also presented in the main manuscript. Please see the following sections that demonstrate the gap in the literature, study significance, and overall problem statement.

The overall problem statement: Lines 63 to 69

The gap in the literature: Lines 71 to 87

The study significance: Lines 99 to 111

2. since periodontitis is a multifactorial pathology with modifiable and non-modifiable risk factors. Several risk factors should be mentioned in the introduction, whether they are local, systemic, linked to the individual, to society, biological, socio-demographic... On the other hand, the system proposed by the authors could have been all the more useful as it would have extracted this information from the EDR.

Response: Thank you for this excellent suggestion. We have now added the information about the risk factors information. We have already extracted information about the risk factors information from the EDR in our previous studies. We have now cited these studies in this manuscript. The rationale for this study was to examine the change in the disease status. In our future studies, we will conduct a time-series analysis to determine the impact of the risk factors on the disease's progression versus healthy controls. We have now clarified this point in our updated manuscript (see lines 56, 94-96 ).

3. I don't understand at all why NLP is necessary since the authors have periodontal charting. Please justifiy and detail.

Response: Thank you for raising this critical question. Even though the completeness of clinician-recorded diagnoses was lower, we utilized them to track the PD change because of the reliability of the information. We utilized the case definition (developed for periodontal surveillance purposes) developed by the American Academy of Periodontology for periodontal charting, not the 2017 diagnostics classification. Since the analyzed data was obtained between 2009 and 2014, the application of the 2017 classification may not be appropriate. Moreover, the case definition does not consider bone loss information for PD diagnoses, while clinician-recorded diagnoses do. We decided to only include the completeness of the periodontal charting automated diagnosis in the paper because, although we did not use it to track PD change, it has other applications. For example, this approach can be utilized to estimate the prevalence of periodontitis in the US, especially given the latest NHANES study estimating US periodontitis prediction model concluded- " prevalence study is not anticipated in the future due to high costs associated with collecting data using prospective study designs". Our paper on this topic has been accepted for the 2023 MEDINFO conference. We have cited this paper in the updated manuscript as well. Moreover, we wanted to compare the completeness of periodontal charting documentation versus clinician-documented diagnosis. We have added this justification and points in the updated manuscript (see lines 229 to 244).

4. Are the authors using the last periodontal classification?

Response: Yes, we have used the latest periodontal classification (case definitions) developed by the AAP to estimate the prevalence of PD. However, since the charting generated diagnoses were not used to track the disease change due to the reasons mentioned above, we removed this information and submitted this manuscript separately to an informatics conference. This paper has been accepted for publication in MEDINFO. We have now added this information in the updated manuscript (see lines 229 to 244).

5. What does mean NLP ? If this article presents a data processing flow, this step must be detailed. Is the NLP based on deep learning ? (in such detail the network structure, dataset training, validation and test) or regular expression (detail the structure of the regexp and its accuracy).

Response: Thank you. We have cited the paper that describes the NLP and other automated programs described in this study. We have also added more content about these approaches in the updated manuscript. However, the objective of this paper was to develop three patient cohorts to study the clinical course of periodontitis. As a result, we cited the paper in this manuscript that provides a detailed description of the computer algorithms, their validation processes, and the Python codes (see lines 127 to 178).

6. Where is the python code ?

Response: Thank you. We have already shared our computer algorithms that automatically diagnose periodontal disease and NLP applications that extract information from free-text clinical notes in the supplemental material. Please see the supplemental materials.

7. I would also have expected the article to focus on the prediction of the evolution of periodontitis from the initial diagnosis and charting.

Response: Thank you for this suggestion. Predicting PD is our next step to pursue. However, as mentioned earlier, the objective of this study was to evaluate the quality of the longitudinal EDR data and develop three patient cohorts to track PD change over time for prediction modeling. Without these rigorous preprocessing steps, developing a prediction model or utilizing this data to study clinical course is impossible. This information is now added in the limitation and future work section (see lines 410 to 421).

8. In the end this article contains very interesting ideas, but from a computer science point of view it is not sufficient (lack of details, lack of implementation) and from a medical point of view it does not bring any tangible element on periodontitis.

Response: This research is multidisciplinary that utilizes technical solutions to address a healthcare problem. Here, we addressed the big challenge of curating a carefully characterized EHR dataset that can be used to develop disease prediction models.As demonstrated in the manuscript, no studies have evaluated the quality of longitudinal EDR data, which is a critical component to be performed before utilizing this data for research including AI models. Moreover, data processing and cleaning take up to 90% of the time. No studies in dentistry have provided these detailed steps to generate patient cohorts. Moreover, existing studies that developed automated PD diagnoses did not provide their codes, making it impossible to reproduce. We have shared all our codes so that other researchers could utilize these resources to process their EDR data, which is usually not a clean data as one would expect. Hence, this study has a high reproducibility potential and content that hasn't been published in the existing literature. Finally, because this study is not computer science focused but more dental science focused, we decided to submit to the Diagnostics journal and not a computer science journal.

Reviewer 3 Report

The paper aims to determine the feasibility of utilizing longitudinal electronic dental record (EDR) data to track change over time in periodontal disease (PD) patients and to generate three patient cohorts: 1)  patients whose disease did not change over time, 2) patients whose PD progressed, and 3) patients whose disease improved over time using informatics approaches. The paper is submitted to the special issue of the diagnostics journal entitled “Advances in Biomedical and Dental Diagnostics Using Artificial Intelligence” but the role of AI is not well discussed. More details regarding the AI methods used should be added. Besides, more results achieved with these AI models should be added. More discussion and analysis of the results should be added.

Abstract:

The paper is submitted to the special issue of the diagnostics journal entitled “Advances in Biomedical and Dental Diagnostics Using Artificial Intelligence” but I cannot find any details in the abstract regarding which artificial intelligence (AI) methods were used and how they were implemented in the manuscript. Also, add the numerical findings achieved with  the  AI model

.

Introduction:

Figure 1 is not cited in the text. Also, the contents of the figure should be well explained.

Please add to the introduction section the role of AI in your proposed system.

Research Methodology

Could you please mention the criteria for the inclusion and exclusion of a participant in the data acquisition process?

Please add more details regarding the methods of natural language processing you used.

Which classifiers were used? Please add more details

More details regarding the implementation of the computer application PD Change Classifier.py should be added

Please add a block diagram discussing the steps of the proposed system.

Results

Please add the confusion matrix and ROC curves to the results.

Please define the evaluation metrics you used.

Please add more discussion on the results of the AI models used.

Discussion and Conclusion

Please add your limitations and future directions.

Author Response

Note: Dear reviewer, I request you to see the attached word document for better visibility.

Response to the Reviewers' Feedback

We want to thank all reviewers and editors for providing their critical feedback and suggestions, which have helped us improve our manuscript tremendously. We want to note that we spent a significant amount of time addressing reviewers' comments especially making the study focus technical to fit the publication in the special issue. We have strengthened the rationale of evaluating the quality of longitudinal data and generating a cohort of patients automatically in the updated manuscript. We have provided detailed descriptions of the two automated algorithms developed in this study to clean and preprocess electronic dental record (EDR) data for research. Finally, we addressed all 24 unique comments (100%) provided by the reviewers (n=4). We have also shared our computer algorithms that automatically diagnose periodontal disease and NLP applications that extract information from free-text clinical notes in the supplemental material. We have also provided access to GitHub and source codes for public use. I hope this increases the research reproducibility. We have made all changes in track change; therefore, please ensure to have the track changes correspond to the accurate line and page numbers. We hope you find these edits helpful and meet your expectations.

Reviewer 3

1. The paper aims to determine the feasibility of utilizing longitudinal electronic dental record (EDR) data to track change over time in periodontal disease (PD) patients and to generate three patient cohorts: 1) patients whose disease did not change over time, 2) patients whose PD progressed, and 3) patients whose disease improved over time using informatics approaches. The paper is submitted to the special issue of the diagnostics journal entitled "Advances in Biomedical and Dental Diagnostics Using Artificial Intelligence" but the role of AI is not well discussed. More details regarding the AI methods used should be added. Besides, more results achieved with these AI models should be added. More discussion and analysis of the results should be added.

Response: Thank you for this suggestion. We agree with you that this paper is not focused on the AI methods themselves. However, we took one step back, determined the quality of the EDR data, and generated a cohort of patients to study the clinical course of periodontitis. As an expert, you must be aware that most of the health information technologies, especially prediction models developed using poor-quality data, may not work in a real-world setting. Therefore, ample literature has recently demonstrated the importance of evaluating the quality of the electronic health record (EHR) data before its intended use for research to avoid the "garbage in, garbage out" situation (cited in the main manuscript). In this study, we attempted to first determine the quality of the longitudinal electronic dental record (EDR) data to examine if this data can be utilized to study periodontal disease (PD). Moreover, the EDR data is messy and automated approaches are needed to utilize the data for research purposes. You may have found our research question simple; however, we developed two complex computational approaches to automate diagnosis. Any researchers worldwide would be able to utilize these open-source computer algorithms to improve the completeness of PD diagnosis documentation. Therefore, we understand that it may not be a fit because there is no involvement of any AI methods; however, it demonstrates very critical preprocessing and data cleaning steps that are required before running any AI models on the data. Finally, we contacted the Editor-in-Chief and Guest Editor of this Special issue, and they seem to be fine with not involving AI methods in this study.

2. Figure 1 is not cited in the text. Also, the contents of the figure should be well explained.

Response: Thank you for this comment. We have now removed the content of periodontal charting information from the manuscript as per other reviewer's suggestion. This is because we did not utilize the PD diagnosis derived from the periodontal charting data to determine the PD change over time. We however kept the results on the observation time related to periodontal charting data to compare the data completeness between periodontal charting and clinician-recorded diagnosis. We have now added this classification in the manuscript. Please see lines 126 to 147.

3. Please add to the introduction section the role of AI in your proposed system.

Response: We have not used any AI models for this research. In this study, we attempted to first determine the quality of the longitudinal electronic dental record (EDR) data to examine if this data can be utilized to study periodontal disease (PD), which is a critical step. Moreover, the EDR data is messy and automated approaches are needed to utilize the data for research purposes. You may have found our research question simple; however, we developed two complex computational approaches to automate diagnosis. Any researchers worldwide would be able to utilize these open-source computer algorithms to improve the completeness of PD diagnosis documentation. Therefore, we understand that it may not be a fit because there is no involvement of any AI methods; however, it demonstrates very critical preprocessing and data cleaning steps that are required before running any AI models on the data. Finally, we contacted the Editor-in-Chief and Guest Editor of this Special issue, and they seem to be fine with not involving AI methods in this study.

4. Could you please mention the criteria for the inclusion and exclusion of a participant in the data acquisition process?

Response: Thank you for this suggestion. We have now added the inclusion and exclusion criteria of the data acquisition process. Please see lines 149 to 159.

5. Please add more details regarding the methods of natural language processing you used.

Response: Thank you. We have added some details about the NLP component. We have also provided a reference to the manuscript that has already been published. This manuscript also has access to the NLP computer program. See lines 160 to 178.

6. Which classifiers were used? Please add more details

Response: As explained earlier, we did not use any AI methods for this project, rather developed a pipeline to process, clean, and organize vast amount of EDR data to run the AI model in our next study. We reached out to the Editor-in-Chief and Guest Editor of this Special Issue, and they seem to be fine with not involving AI methods in this study.

7. More details regarding the implementation of the computer application PD Change Classifier.py should be added

Response: Thank you. We have added more information about the implementation of the PD Change Classifier.py in the updated manuscript (see lines 200 to 208).

8. Please add a block diagram discussing the steps of the proposed system.

Response: Thank you for this suggestion. We have added a block diagram in the updated manuscript. Please see lines 160 to 178.

9. Please add the confusion matrix and ROC curves to the results.

Response: Thank you. We have added further information about the confusion matrix and ROC curve. Please see lines 267 to 281 and Tables 1 and 2.

10. Please define the evaluation metrics you used.

Response: Thank you for this suggestion. Detailed information about the evaluation metrics have been added in the updated manuscript. Please see lines 267 to 281.

11. Please add more discussion on the results of the AI models used.

Response: As mentioned earlier, we did not use any AI methods for this project, rather developed a pipeline to process, clean, and organize vast amount of EDR data to run the AI model in our next study. We reached out to the Editor-in-Chief and Guest Editor of this Special Issue, and they seem to be fine with not involving AI methods in this study.

12. Please add your limitations and future directions.

Response: Thank you for this suggestion. We have added limitations and future directions in the updated manuscript. Please see lines 411 to 421.

Round 2

Reviewer 3 Report

The authors have enhanced the quality of the manuscript, but there are still some comments to be addressed.

The authors claimed that they did not use AI, however they are using natural language processing (NLP) which is an AI technique.

How did the author perform classification in order to calculate the accuracy, sensitivity, specificity metrics?